# Position: Causality Is Key to Understand and Balance Multiple Goals in Trustworthy ML and Foundation Models

## Abstract

Ensuring trustworthiness in machine learning (ML) systems is crucial as they become increasingly embedded in high-stakes domains. This paper advocates for the integration of causal methods into machine learning to navigate the trade-offs among key principles of trustworthy ML, including fairness, privacy, robustness, accuracy, and explainability. While these objectives should ideally be satisfied simultaneously, they are often addressed in isolation, leading to conflicts and suboptimal solutions. Drawing on existing applications of causality in ML that successfully align goals such as fairness and accuracy or privacy and robustness, **this position paper argues that a causal approach is essential for balancing multiple competing objectives in both trustworthy ML and foundation models.** Beyond highlighting these trade-offs, we examine how causality can be practically integrated into ML and foundation models, offering solutions to enhance their reliability and interpretability. Finally, we discuss the challenges, limitations, and opportunities in adopting causal frameworks, paving the way for more accountable and ethically sound AI systems.

## 1. Introduction

In recent years, machine learning (ML) has made remarkable strides, driving breakthroughs in natural language processing (Achiam et al., 2023), computer vision (Brooks et al., 2024), and decision-making systems (Jia et al., 2024). These advancements have led to widespread adoption across diverse domains, including healthcare (Singhal et al., 2025), finance (Lee et al., 2024), education (Team et al., 2024), and social media (Bashiri & Kowsari, 2024),

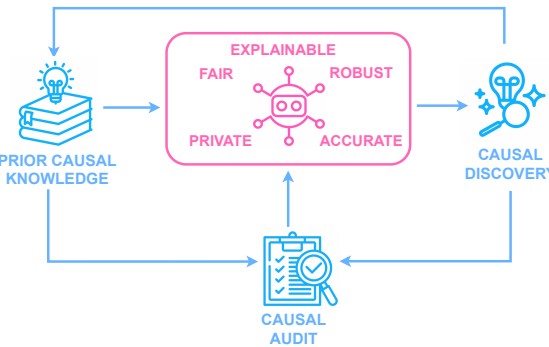

Figure 1: Causal Trustworthy ML Cycle: Causal ML can leverage existing knowledge and causal auditing to enhance different components of trustworthiness: explainability, fairness, privacy, and accuracy while simultaneously advancing understanding through causal discovery.

where ML models now play a crucial role in diagnostics, algorithmic trading, personalized learning, and content recommendation.

Given their soaring influence, it has become a global priority to ensure ethical and trustworthy ML systems. Many international regulations and frameworks (European Commission, 2021; OECD, 2019; Group of Twenty (G20), 2019; Infocomm Media Development Authority, 2020) seek to establish guidelines for fairness, explainability, robustness, and privacy protection. For the scope of our paper, we are aware of different definitions of trustworthiness, but will focus on five core dimensions that are both widely recognized and directly relevant to causal reasoning: fairness, privacy, robustness, explainability, and accuracy. We will introduce these dimensions and highlight their trade-offs and intersections below.

**Fairness.** Fairness in ML refers to the principle that systems should make unbiased decisions that do not discriminate against individuals or groups based on sensitive attributes such as race, gender, or socioeconomic status. ML systems have been shown to rely heavily on biased data, amplifying existing biases and leading to unequal outcomes (COMPAS, 2020). These systems often exhibit reduced accuracy for minority or underrepresented groups, further exacerbating disparities (Buolamwini &

[1]Anonymous Institution, Anonymous City, Anonymous Region, Anonymous Country. Correspondence to: Anonymous Author <anon.email@domain.com>.

Gebru, 2018). Given the speed and scale of ML-enabled decisions, ensuring fairness is essential to prevent perpetuating and exacerbating societal inequalities at an unprecedented scale.

**Privacy.** Privacy in ML emphasizes the protection of individuals' sensitive and personal data. It has been shown that even after removing identifiers such as names, information can still leak, and individuals can be reidentified through indirect attributes and data triangulation(Sweeney, 2000; Narayanan & Shmatikov, 2008; Ohm, 2010; Dwork, 2006). Additionally, sensitive information can be reconstructed from gradients during model training if data is not handled privately (Zhu et al., 2019; Geiping et al., 2020; Aono et al., 2017; Fredrikson et al., 2015). Privacy is crucial for ensuring compliance with data protection laws and safeguarding human rights. It also fosters trust for individuals to be more willing to contribute their data for model training if their safety and privacy were ensured.

**Robustness.** Robustness refers to the system's ability to perform reliably under varying conditions, including adversarial attacks, noisy inputs, or distributional shifts. For example, models often underperform when faced with distribution shifts, such as changes in data characteristics between training and deployment environments (Hendrycks & Dietterich, 2019; Recht et al., 2019; Ovadia et al., 2019). Additionally, human-undetectable noise added to images can cause models to make incorrect predictions, highlighting their vulnerability (Szegedy et al., 2014; Goodfellow et al., 2015). Robustness is critical to ensuring the safety and reliability of AI systems, particularly in high-stakes applications such as healthcare and autonomous driving.

**Explainability.** Explainability refers to the ability of AI systems to provide clear and understandable reasoning behind their decisions or predictions. Deep neural networks (DNNs), often referred to as "black boxes," are inherently complex and difficult to interpret, making them hard to audit and assess for fairness or correctness (Lipton, 2018; Doshi-Velez & Kim, 2017; Rudin, 2019). Explainability is closely tied to accountability, as it enables stakeholders to evaluate and challenge AI outputs when necessary. Furthermore, regulations such as the GDPR emphasize the "right to explanation," which requires that individuals be informed about and understand how automated decisions affecting them are made (European Comission, 2016).

**Trade-offs and Intersections.** The trustworthy ML landscape involves complex trade-offs and interdependencies between key objectives such as fairness, privacy, accuracy, robustness, and explainability. Improving one aspect often comes at the expense of another, such as the trade-off between **privacy** and accuracy in differential privacy, where noise added to protect data reduces model accuracy (Xu et al., 2017; Carvalho et al., 2023). Similarly,

achieving **fairness** frequently requires sacrificing predictive performance or resolving conflicts between competing fairness notions, such as demographic parity and equalized odds (Friedler et al., 2021; Kim et al., 2020). Trade-offs also arise in **explainability** and accuracy, as complex models like DNNs excel in performance but lack interpretability. Meanwhile, the relationship between fairness and privacy is nuanced, with evidence showing they can either conflict, as noise may lead to disparate outcomes, or complement each other by reducing bias (Pujol et al., 2020; Dwork et al., 2011).

**Causality.** One of the most influential causal frameworks is Pearl's structural causal models (SCMs), which provide a systematic approach to reasoning about causality and integrating it into machine learning (Pearl, 2009b). This framework defines causality as the relationship between the variables where a change in one variable (*the cause*) directly leads to a change in another variable (*the effect*). It establishes a directional and often mechanistic link, distinguishing relationships arising from mere correlations.

A key component of Pearl's framework is the use of directed acyclic graphs (DAGs) and do-calculus, which offer a structured representation of causal dependencies and a formal method for performing causal inference. A causal DAG, denoted as $\mathcal{G} = (\mathbf{V}, \mathcal{E})$, consists of a set of nodes $\mathbf{V}$ representing random variables and directed edges $\mathcal{E}$ encoding causal relationships among the variables.

Unlike correlation-based approaches, causality provides a framework for disentangling the underlying mechanisms that drive observed phenomena, offering a deeper interpretation of data. Causal frameworks have been successfully applied to audit and mitigate fairness (Kim et al., 2021; Kilbertus et al., 2017; Loftus et al., 2018) and to improve robustness (Schölkopf, 2022). The research about the connection between causality and privacy is still very limited, but some emerging studies show potential for applications (Tschantz et al., 2020). Finally, explainability is one of the core features of causality and comes pre-packaged with the causal framework. Despite the promising applications of causality for individual requirements of trustworthy AI, the potential to use causality to reconcile individual requirements of trustworthy ML remains largely underexplored.

**Position.** Despite significant advancements in research on individual dimensions of trustworthy ML such as fairness, privacy, and explainability—there is a notable lack of efforts to integrate these dimensions into a cohesive and unified framework. Each ethical principle addresses distinct challenges, yet their interplay often involves intricate trade-offs, particularly concerning model performance metrics such as accuracy. For example, mitigating fairness-related biases may require adjustments that compromise predictive

precision, while enhancing explainability can impose constraints on model complexity. We argue that systematically addressing these trade-offs is a critical step toward developing AI systems that are both ethically sound and operationally efficient. While causality has been applied to address individual challenges such as fairness or interpretability, its potential to address the intersection of these challenges has largely been overlooked (see Appendix A for a detailed review). **In this position paper, we argue that integrating causality into ML and foundation models offers a way to balance multiple competing objectives of trustworthy AI.**

The structure of our paper is as follows. Section 2 analyzes how causality can reconcile multiple dimensions of trustworthy ML and explores how it can be integrated. Section 3 discusses how foundation models amplify existing ML trade-offs and introduce new challenges, for which we argue that causality provides a principled approach to overcoming these issues, and propose strategies for integrating causal reasoning into foundation models at different development stages. Section 4 covers limitations in applying causality to ML and foundation models and proposes future research directions, and Section 5 includes alternative views. Finally, Section 6 suggests key steps for advancing causality in ML and foundation models.

## 2. Causality for Trustworthy ML

Trustworthy ML involves inherent trade-offs between core objectives such as accuracy, fairness, robustness, privacy, and explainability. Inevitable trade-offs can exist between accuracy and other objectives, fairness and privacy, and conflicting fairness notions. However, some other goals may reinforce each other, such as explainability aiding fairness assessment, and privacy enhancing robustness (Dwork & Lei, 2009; Hopkins et al., 2022).

Causality provides a principled approach to navigating these trade-offs by explicitly modeling data-generating processes and clarifying assumptions. This section first explores causal formulations for these trade-offs, and then introduce how causality can mitigate these tensions and support a more balanced approach to trustworthy ML.

### 2.1. Causality for Trade-offs in Trustworthy ML

In this section, we examine key trade-offs in trustworthy ML and illustrate how causal approaches can help reconcile these competing objectives.

**Privacy vs. Accuracy.** The differential privacy approach relies on adding noise to the data which is controlled by the parameter $\epsilon$ (the smaller value of $\epsilon$ corresponds to more noise, while the larger value indicates less noise and less privacy). Naturally, it hurts the accuracy of an algorithm

learned on the privatized data. It is yet unknown how to avoid this fundamental trade-off between data protection and the utility of the data (Xu et al., 2017; Carvalho et al., 2023). One of ways how causality can inform privacy is provided by (Tschantz et al., 2020). The authors define privacy violations as causal effects, emphasizing that private information is leaked when an adversary can infer sensitive attributes from observable data due to causal pathways. Therefore, causal models can help identify, quantify, and mitigate such risks, offering a more systematic alternative to heuristic-based privacy measures.

By aligning privacy interventions with causal relationships, models can obscure sensitive attributes (e.g., sex, race) while preserving meaningful data dependencies, reducing the negative impact on accuracy. For example, causal graphs ensure that interdependent variables (e.g., age and education) are randomized together to avoid unrealistic combinations (e.g., "Age: 5; Education: Bachelor"). Preventing such inconsistencies not only improves accuracy but also reduces the likelihood of adversaries exploiting obfuscation patterns, enhancing overall privacy protection.

**Fairness vs. Accuracy.** Most of the statistical fairness literature focuses on improving fairness metrics while preserving accuracy as much as possible (Feldman et al., 2015; Calders & Verwer, 2010; Wei & Niethammer, 2022; Wang et al., 2021a). However, fairness often comes at the cost of reduced accuracy, as mitigating bias may require either obscuring predictive features that also contribute to discrimination or constraining model predictions within fairness-imposed boundaries (Pinzón et al., 2022; Cooper et al., 2021; Zliobaite, 2015; Zhao & Gordon, 2022).

A key issue is that many fairness-accuracy trade-offs arise from addressing correlations rather than causal relationships. Causal models can resolve these tensions by distinguishing legitimate predictive factors from spurious discriminatory pathways. By disentangling the direct and indirect effects of sensitive attributes on outcomes, causal interventions can mitigate unfair biases without sacrificing accuracy. For instance, counterfactual fairness ensures that individuals receive the same prediction regardless of their sensitive attributes in a counterfactual world where those attributes are altered (Kusner et al., 2017).

A compelling example comes from the COMPAS dataset, where Black defendants were more likely to be classified as high-risk for recidivism. Traditional statistical debiasing approaches treat race as a direct cause of the risk score, but a causal analysis reveals that increased recidivism risk is confounded by heightened policing in predominantly Black neighborhoods. By explicitly modeling this causal structure, fairness-enhancing interventions can adjust for the effect of over-policing, ensuring that predictions reflect true recidivism risk rather than biased enforce-

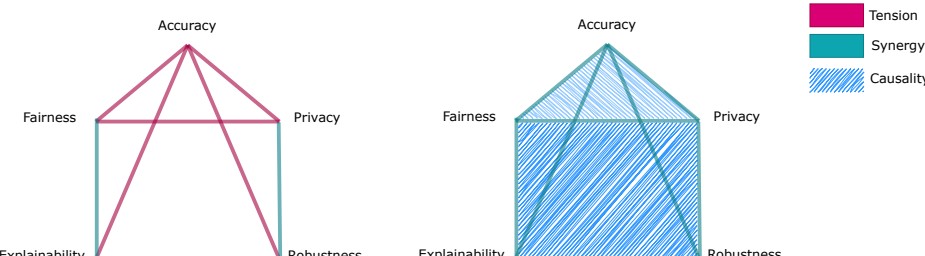

Figure 2: While trustworthy AI involves inherent trade-offs between its key components, causality can help mitigate these tensions and enhance synergies.

ment patterns. This results in a more accurate and fairer risk assessment (Chiappa, 2019; Zafar et al., 2017; Zhang & Bareinboim, 2018).

**Conflicting Notions of Fairness.** Fairness in ML is often constrained by conflicting definitions and measurement approaches. Friedler et al. (2021) highlight the fundamental tension between the "what you see is what you get" and "we are all equal" worldviews—where the former accepts disparities based on observed merit, while the latter seeks to correct historical inequalities. Causal graphs can crisply formulate different notions of fairness (Nabi & Shpitser, 2018; Chen et al., 2024b), thus enabling feasible mitigation via path-specific causal effects (Avin et al., 2005).

Kim et al. (2020) formalize fairness conflicts using the fairness-confusion tensor, showing that notions like demographic parity and equalized odds impose incompatible constraints. The causal approach mitigates these conflicts by focusing on fairness as a property of causal pathways rather than statistical dependencies (Rahmattalabi & Xiang, 2022). This allows for greater flexibility in aligning fairness interventions with real-world causal mechanisms, allowing better-informed choice of fairness metric.

**Robustness vs. Accuracy.** The trade-off between generalizability and accuracy is rooted in the observation that models trained to achieve high accuracy on a specific dataset often overfit to the peculiarities of that distribution. This overfitting compromises their ability to generalize to new, unseen distributions (Schölkopf, 2022). On the contrary, causal models focus on invariant relationships that hold across different environments, making them robust to distribution shifts. This robustness enhances the model's ability to generalize to unseen data, improving accuracy in diverse settings. For example, causal representation learning disentangles stable causal factors, allowing the model to maintain performance when data distributions change. Moreover, Richens & Everitt (2024) prove that robust agents implicitly learn causal world models, further emphasizing the intrinsic interdependency between robustness and causality.

**Explainability vs. Accuracy.** Many complex algorithms, such as deep neural networks (DNN) or random forest (RF), have impressive predictive power but provide "blackbox" solutions that are hard to question or evaluate (London, 2019; van der Veer et al., 2021). Causal models offer inherently interpretable structures by quantifying the contribution of each input feature to the output, providing clear, human-understandable explanations. Causal recourse further enhances explainabilit by offering actionable recommendations for individuals affected by model decisions, helping them achieve a more favorable outcome (Karimi et al., 2021).

**Fairness vs. Explainability.** A particularly powerful approach within causal explainability is counterfactual explanations, which help users understand model decisions by asking "what if" questions. Counterfactual methods generate alternative scenarios where certain features are changed while keeping others constant, allowing for a direct assessment of how specific inputs influence predictions (Wachter et al., 2017; Karimi et al., 2020). Counterfactual explanations are particularly useful for fairness auditing as they can help identify why certain groups are adversely affected and guide corrective measures.

**Privacy vs. Robustness.** Adding noise without considering the data structure or causal relationships can obscure meaningful patterns and introduce spurious correlations. This indiscriminate noise can make models less robust to unseen data, particularly under distribution shifts.

In contrast, causal models inherently emphasize invariant relationships—patterns that are stable across various data distributions. Noise that disrupts non-causal relationships or spurious correlations can further enhance the robustness of these models to shifts in data. Finally, some results show, that causal models provide stronger guarantees for adversarial robustness with lower epsilon in differential privacy, thus allowing for lesser negative impact on accuracy (Tople et al., 2020).

**Privacy vs. Fairness.** Privacy mechanisms, such as noise addition, can disproportionately impact minority groups, leading to fairness concerns. Differentially Private Stochastic Gradient Descent (DP-SGD), for example, has been shown to degrade model accuracy more severely for underrepresented groups, exacerbating fairness disparities (Bagdasaryan et al., 2019). However, Causal models can guide privacy interventions by ensuring that noise is applied in ways that do not disrupt fairness-critical relationships. For instance, a causal graph can reveal which features or pathways should be preserved to maintain fairness while protecting privacy.

**Prediction Accuracy vs. Intervention Accuracy.** One of the key advantages of the causal framework is its ability to support not just prediction but also intervention (Hernán & Robins, 2020; Schölkopf, 2022). While predictive models are sufficient in some domains, many high-stakes applications—such as healthcare, policy-making, and personalized treatment—require actionable interventions. In these settings, understanding causal relationships is essential, as the objective is not only to predict outcomes but also to influence them.

### 2.2. Integrating Causality into ML

Integrating causality into ML enables models to move beyond pattern recognition and learn underlying mechanisms governing data. This section explores different approaches to causal ML, ranging from explicitly constrained models that follow predefined causal structures to methods that infer causal relationships from data.

**Causally Constrained ML (CCML).** CCML refers to approaches that *explicitly* incorporate causal relationships into model training or inference as constraints or guiding principles. Given a causal graph $\mathcal{G} = (\mathbf{V}, \mathbf{E})$, where $\mathbf{V}$ represents variables and $\mathbf{E}$ denotes directed edges encoding causal relationships, the goal is to ensure that the learned model $f : \mathbf{X} \to Y$ adheres to the causal structure encoded in $\mathcal{G}$ (Berrevoets et al., 2024; Zinati et al., 2024; Afonja et al., 2024; Schölkopf et al., 2016).

**Invariant feature learning (IFL).** IFL relies on discovered *implicit* or latent causal features and structures. The task of Invariant Feature Learning (IFL) is to identify features of the data $\mathbf{X}$ that are predictive of the target $Y$ across a range of environments $\mathcal{E}$. From a causal perspective, the causal parents $\mathrm{Pa}(Y)$ are always predictive of $Y$ under any interventional distribution (Kaddour et al., 2022). IFL can be achieved by regularizing the model or providing causal training data that is free of confounding.

**Disentangled VAEs.** VAEs aim to decompose the data $\mathbf{X}$ into disentangled latent factors $\mathcal{Z}$ that correspond to distinct underlying generative causes (Burgess et al., 2018). It

can be combined with interventional experiments in mechanistic interpretability that involve "switching off" specific neurons or circuits to gain knowledge about causal workings of the complex model (Leeb et al., 2022). Causality is also used to audit models for fairness (Cornacchia et al., 2023; Byun et al., 2024) or robustness (Drenkow et al., 2024), providing insights into how decisions are influenced by sensitive variables and under distribution shifts.

**Double Machine Learning (DML).** DML provides another causal approach by leveraging modern ML techniques for estimating high-dimensional nuisance parameters while preserving statistical guarantees in causal inference (Chernozhukov et al., 2018). DML decomposes the estimation problem into two stages: (1) predicting confounders using ML models and (2) estimating the causal effect using residualized outcomes.

**Causal Discovery.** Finally, ML can be leveraged for causal inference or to discover causal knowledge from observational data. For instance, methods for causal discovery use statistical patterns to infer causal relationships, with notable examples including (Spirtes et al., 2000; Shimizu et al., 2006; Janzing & Schölkopf, 2010; Peters et al., 2011; Hauser & Bühlmann, 2012; Le et al., 2016).

All of the above forms a causal ML cycle (Figure 1) in which ML is enhanced by causal knowledge, controlled by causal tools, and finally contributes to enriching scientific knowledge. We include a supplementary introduction to causality and causal ML in Appendices C and D.

## 3. Causality for Trustworthy Foundation Models

Foundation models, including state-of-the-art multimodal systems like Large Language Models (LLMs) and vision-language models, have demonstrated exceptional capabilities across diverse tasks (Achiam et al., 2023; Team et al., 2023; Radford et al., 2023; Brooks et al., 2024). However, their reliability remains a concern due to issues like spurious correlations, hallucinations, and unequal representation. Trade-offs and causality in trustworthy foundation models remain underexplored as compared to traditional ML. In this section, we explore the potential for causality to improve fairness, explainability, privacy, and robustness in foundation models following slightly different taxonomy than in the previous section due to their unique challenges.

### 3.1. Dimensions of Trustworthy Foundation Models

In this section, we examine foundation model-specific trade-offs between key dimensions of trustworthy AI and illustrate how causal approaches can soften those tensions.

**Fairness vs. Accuracy.** Causal frameworks have be-

come integral to fairness interventions in LLMs by identifying and mitigating pathways that lead to unfair predictions (Madhavan et al., 2023a; Cotta & Maddison, 2024). Counterfactual fairness ensures that sensitive attributes (e.g., gender, race) do not causally influence outcomes. For example, in job recommendation systems, counterfactual fairness guarantees identical recommendations for equally qualified candidates regardless of their gender (Madhavan et al., 2023a). Methods like causal disentanglement isolate sensitive features from output-relevant causal factors, ensuring that spurious correlations, such as gender biases in job roles, do not propagate through the model (Zhou et al., 2023a; Chen et al., 2024a). SCMs further enable fairness-aware fine-tuning by disentangling causal effects. However, striving for diversity has been shown to introduce non-factual output in text-to-image models. In early 2024, Google's AI tool, Gemini, faced criticism for generating historically inaccurate images, such as depicting America's Founding Fathers as Black individuals and Nazis as racially diverse (Vincent, 2024). Here, causality could help distinguish historically impossible scenarios from desirable diversity, ensuring both fairness and factual integrity in AI-generated content. Mode collapse is another foundation model-specific fairness issue where models generate overly generic outputs, reducing diversity and disproportionately omitting minority group representations. Causal modeling can potentially help preserve minority information by explicitly capturing causal relationships, preventing spurious correlations from erasing underrepresented patterns.

**Robustness vs. Accuracy.** Causal frameworks address robustness by training models to rely on invariant causal relationships while penalizing reliance on dataset-specific spurious features (Wu et al., 2024). For instance, instead of associating "doctor" with "male," causal invariance enforces reliance on task-relevant features like medical terminology (Zhou et al., 2023a). Causal regularization further discourages attention to non-causal patterns during inference achieving better accuracy and robustness.

**Privacy vs. Attribution.** Causal approaches to privacy focus on detecting and severing pathways involving personally identifiable information (PII) in LLMs. Causal obfuscation uses SCMs to identify and block sensitive pathways (e.g., names, locations) during training or inference (Chu et al., 2024). Unlike traditional privacy-preserving mechanisms that indiscriminately apply noise or randomization, it ensures that only privacy-sensitive dependencies are removed, preserving essential predictive relationships.

Beyond conventional privacy concerns, attribution and memorization pose significant challenges in foundation models. Attribution is crucial in determining whether specific data—such as an artist's work—has contributed to the training of a model, enabling rightful recognition and com-

pensation. Memorization, on the other hand, prevents effective data removal, meaning that once a copyrighted work is embedded into a model, it becomes difficult to erase upon request. Causal auditing (Sharkey et al., 2024) potentially offers a principled way to address these challenges by providing a structured framework to verify whether a given dataset—such as an artist's work—has influenced the model's outputs. Unlike statistical correlation-based methods, which may falsely associate stylistic elements with broader art movements, causal auditing can disentangle direct influences from broader historical trends, ensuring that attribution is based on actual data contributions rather than incidental similarities.

**Explainability vs. Capability.** Although foundation models demonstrate remarkable capabilities in various tasks, their outputs often lack interpretability, making it difficult to understand or explain their reasoning. Causal models can help quantify how much each input feature contributes to a specific output, providing a clear and interpretable explanation. By modeling causal chains, we can explain how different stages of the LLM (e.g., embedding, attention layers, output logits) interact to produce a final decision (Bagheri et al., 2024). This creates a step-by-step explanation of the model's reasoning process. Another domain that is related to causality is mechanistic interpretability. Mechanistic interpretability seeks to decode the inner workings of LLMs by analyzing their architecture, weights, and activation patterns (Conmy et al., 2023). Causality enhances this understanding by identifying cause-effect relationships within these mechanisms. Causality can identify specific pathways in neural circuits that contribute to certain outputs (Palit et al., 2023; Parekh et al., 2024). For example, specific neurons or attention heads affect token predictions, revealing the factors driving outputs.

### 3.2. Integrating Causality in Foundation Models

This section delves into practical applications of causality in FMs across three key stages: pre-training, post-training, and auditing. We conclude with a discussion of the practical advantages and limitations of the proposed approaches.

**Pre-training: Causal data augmentation.** Synthetic datasets with explicit causal structures, such as counterfactual examples or causal-transformable text data, can be used to augment training data. Counterfactual data augmentation introduces scenarios where causal relationships differ from spurious correlations, helping models learn true causal dependencies instead of misleading patterns (Webster et al., 2020; Chen et al., 2022).

**Pre-training: Causal Representation Learning.** By disentangling causal factors from non-causal ones, models can learn representations that separate meaningful causal features from irrelevant associations. Techniques such as

causal embedding methods (Rajendran et al., 2024; Jiang et al., 2024), which can use training data annotated with causal labels, can guide models in identifying and prioritizing true causal relationships. This has been shown to reduce reliance on spurious correlations, such as gender-biased occupational associations (Zhou et al., 2023b).

**Pre-training: Entity interventions.** SCMs can be used to intervene on specific entities (e.g., replacing "Joe Biden" with "ENTITY-A") during pre-training (Wang et al., 2023), thus reducing entity-based spurious associations while preserving causal relationships in the data.

**Pre-training: Loss function.** Modifying the pre-training loss function to penalize reliance on confounders can help align models with causal principles. For instance, fine-tuning models on embeddings pre-trained with debiased token representations has shown promise for causal learning (Kaneko & Bollegala, 2021; Guo et al., 2022; He et al., 2022; Wang et al., 2023).

**Post-training: Fine-tuning.** Fine-tuning on datasets specifically designed to highlight causal reasoning (e.g., datasets emphasizing cause-effect linguistic patterns) ensures that models learn causal-invariant patterns. Further, counterfactual data samples can also improve the fine-tuning. Synthetic counterfactual examples improve the model's robustness to spurious correlations, similar to pre-training, but with better sample size efficiency. Frameworks like DISCO (Chen et al., 2022) generate diverse counterfactuals during fine-tuning to enhance OOD generalization for downstream tasks. Causally Fair Language Models (CFL) (Madhavan et al., 2023b) use SCM-based regularization to detoxify outputs or enforce demographically neutral generation during post-training. Wang & Culotta (2020) use causal reasoning to separate genuine from spurious correlations by computing controlled direct effects, ensuring robust performance.

**Post-training: Alignment.** RLHF can be adapted to include causal interventions, allowing feedback to act as instrumental variables that correct biased model behavior. Causality-Aware Alignment (CAA) (Xia et al., 2024) incorporates causal interventions to reduce demographic stereotypes during fine-tuning with alignment objectives. Extending RLHF with causal alignment to support dynamic, context-sensitive interventions could help address biases that evolve. Integrating causal reasoning into the reward model's decision-making process, by critiquing the output of LLM using a reward model or a mixture of reward models that control for specific confounders or spurious correlations can potentially improve the downstream reasoning abilities potentially mitigating hallucinations.

**Auditing and Evaluation.** Causality provides a structured framework for auditing privacy risks by identifying whether sensitive user data contributes to model outputs. This is particularly important for privacy regulations like GDPR's "right to be forgotten" (European Comission, 2016), where users can request their data to be removed from an AI system. However, verifying whether an LLM has truly forgotten a user's data is a complex challenge, as models can memorize training information in ways that are difficult to detect through standard evaluation metrics. One key approach in privacy auditing is using causal attribution, which assesses whether a specific data point influenced a given output. By using do-calculus, privacy auditors can evaluate how an output changes when a particular data source is removed. This enables a principled test of whether an LLM has truly forgotten a user's data.

*Practical Considerations.* In supervised fine-tuning and alignment, the downstream task and its causal relationships are often known, allowing for more targeted interventions on confounding variables and even the collection of task-specific data to refine causal structures. Additionally, since post-training typically requires less data than pre-training, integrating causal insights becomes more feasible. Pre-training offers the advantage of learning broad representations from diverse data, but it is difficult to enforce causal constraints due to the lack of explicit task definitions and causal structures. Auditing is particularly useful for detecting biases, ensuring fairness, and validating robustness in real-world scenarios. Unlike pre-training and fine-tuning, auditing does not require modifying the training pipeline, making it a cost-effective way to introduce causal reasoning retrospectively.

## 4. Challenges and Opportunities

Despite its advantages, there are many challenges when applying causality to trustworthy ML, including reliance on strong causal assumptions and limited availability of *a priori* causal knowledge, particularly in the form of DAGs. Foundation models bring further complications due to their scale, high-dimensional data, and the difficulty of validating causal structures. We outline key obstacles in integrating causality into ML and foundation models and suggest strategies to overcome them.

**Availability of Causal Knowledge.** A major challenge in causal ML is the limited availability of causal knowledge, particularly in the form of DAGs. Expert-constructed DAGs may suffer from subjectivity and scalability issues, while ML-based causal discovery is constrained by identifiability assumptions and noise sensitivity. However, recent hybrid approaches combining classical causal discovery with LLM-based reasoning offer promising solutions.

**Causal Transportability.** Scientific knowledge often lacks direct applicability across different populations, making

causal transportability essential. Pearl and Bareinboim's DAG-based framework adjusts causal knowledge for new settings using targeted data collection (Pearl & Bareinboim, 2011b; Bareinboim & Pearl, 2014; Pearl & Bareinboim, 2011a). Building on this, Binkyte et al. (2024) propose an expectation-maximization (EM) approach to adapt causal knowledge for target demographic applications.

**Potentially Unresolvable Tensions.** Not all tensions in trustworthy AI can always be fully resolved. For instance, stronger privacy protections often reduce model utility (Dwork et al., 2014; Bassily et al., 2014). Similarly, explainability may sometimes come at the cost of accuracy, and robustness can conflict with fairness in certain scenarios. However, causality provides a structured approach to evaluating these trade-offs, making it possible to quantify their impact and identify cases where full reconciliation is not feasible. Importantly, it is crucial to be transparent about these limitations, as this fosters societal trust, promotes accountability, and enables more informed decision-making in AI development.

**Challenges in Causal Foundation Models.** One foundation model-specific challenge is concept superposition, particularly in LLMs, where multiple meanings are entangled within a single representation, complicating causal reasoning (Elhage et al., 2022). Vision models exhibit this issue to a lesser extent due to their structured data formats. Being aware of superposition is imporant for effectively integrating causality.

Another challenge is the lack of high-quality causal data. Training foundation models with causal reasoning requires datasets annotated with explicit causal structures or interventional data, which are scarce and expensive to produce. Scalable methods for generating synthetic causal datasets show a promising direction. Alternatively, focusing on post-training methods allows causal interventions in a more data-efficient way.

Additionally, the computational complexity of integrating causal reasoning into foundation models poses a significant challenge. For fine-tuning, low-rank adaptation methods such as LoRA can be employed to reduce the number of learnable parameters, making causal integration more efficient without compromising performance (Hu et al., 2021).

## 5. Alternative View

Some may argue that different domains prioritize different requirements for trustworthy ML, and there is no need to reconcile them. However, this perspective is unlikely to hold universally, as most real-world applications intersect with multiple ethical and trustworthy ML principles, such as fairness, privacy, and robustness, which must be balanced to ensure reliable outcomes.

Another perspective suggests that causal properties can emerge spontaneously by training larger models on vast amounts of data. While this is possible, it is not guaranteed, and more importantly, it provides no control over whether or how these properties arise. In contrast, much of the scientific causal knowledge already exists, and finding ways to integrate this knowledge with machine learning models offers a more resource-efficient, reliable, and explainable pathway to achieving trustworthy ML.

## 6. Conclusion and Call for Action

Causal models offer a principled approach to trustworthy AI by prioritizing relationships that are causally justified and invariant across contexts. This approach reduces tensions between competing objectives and can enhance multiple dimensions—privacy, accuracy, fairness, explainability, and robustness—simultaneously, creating models that are not only ethically sound but also practically effective.

To further advance trustworthy ML foundation models, we emphasize the need for the following actions:

*Incorporate Trade-off Awareness in Model Design*: Ensure that foundation models are developed with explicit consideration of trade-offs between key trustworthy AI dimensions—fairness, privacy, robustness, explainability, and accuracy.

*Leverage Causality to Resolve or Soften Trade-offs*: Where possible, integrate causal reasoning to disentangle competing objectives and mitigate conflicts.

*Develop Scalable Methods for Causal Data Integration*: Encourage the development of algorithms and pipelines to integrate causal knowledge into foundation models at scale.

*Create and Share High-Quality Causal Datasets*: Foster initiatives to curate, annotate, and share datasets with explicit causal annotations or interventional information.

*Advance Causal Discovery Techniques*: Invest in research to improve causal discovery algorithms. Hybrid approaches combining classical methods with LLM-based contextual reasoning show a promising direction.

*Benchmark and Evaluate Causal Models*: Establish evaluation frameworks that assess the ability of causal models to balance trade-offs effectively and provide transparent justifications for their decisions in high-stakes domains.

All these advancements are crucial for expanding the application of causality in ML and foundation models, paving the way for more balanced and trustworthy AI solutions.

## 7. Impact Statement

This paper advocates for integrating causality into foundation models to enhance fairness, privacy, robustness, and explainability. By reducing reliance on spurious correlations and improving decision-making, causal methods can make AI systems more reliable, transparent, and aligned with human values—especially in high-stakes domains like healthcare, law, and finance. Adopting causality-driven AI has the potential to improve trust, regulatory compliance, and ethical governance, ultimately contributing to a more fair, transparent, and socially beneficial technological landscape.

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

## A. Related work

The literature on trade-offs in ethical AI emphasizes the inherent tensions and competing objectives involved in designing and deploying AI systems that align with ethical principles. The works of Sanderson et al. (2024); Whittlestone et al. (2019); Katirai & Nagato (2024) explore frameworks for balancing fairness, accuracy, and other conflicting priorities. Kemmerzell & Schreiner (2024) explore trade-offs between robustness, accuracy, fairness, and privacy, and suggest data augmentation techniques that minimize the trade-offs.

The surveys on the use of causality for trustworthy AI (Ganguly et al., 2023; Rawal et al., 2024; Liu et al., 2023) provide a comprehensive overview of the existing use cases. However, they do not discuss the need for causality explicitly and do not focus on the role of causality in alleviating tensions in trustworthy AI. Notably, only Ganguly et al. (2023) overview the use of causality in privacy and exclusively focuses on the adversarial robustness through generalization. Discussions on causality for ethical AI focusing on challenges and applications are provided in (Rahmattalabi & Xiang, 2022; Vallverdú, 2024).

Several works discuss the benefits of use of causality for one or two of the aspects of trustworthy AI. Discussion on the need for causality for fairness can be found in the work of (Binkytė et al., 2022; Plecko & Bareinboim, 2024; Makhlouf et al., 2024). The study by (Wang et al., 2021b; Ehyaei et al., 2023) explored causality to enhance fairness and robustness.

## B. Causality. Frameworks and Definitions

The field of statistical causality encompasses a diverse range of theories and approaches that often complement or compete with each other, rather than forming a unified framework. Researchers have likened the current state of statistical causality to "probability theory before Kolmogorov" (Dawid, 2015). In practice, the application of statistical causality typically involves combining tools and methods from multiple frameworks. This section provides an overview of the existing landscape, highlighting key theories and definitions. Most approaches conceptualize causation either as a relationship revealed through linear regression, grounded in the notion of real or hypothetical interventions, or requiring a mechanistic understanding of the underlying processes (Berzuini et al., 2012). In this work, we primarily rely on the structural probabilistic models framework (Pearl, 2009a) and the potential outcomes framework (Rubin, 2005). Below, we provide an overview of these frameworks and briefly touch on other approaches to causality. For technical definitions of relevant causal concepts, refer to the Technical Preliminaries C.

### B.1. Potential Outcome Framework

The potential outcomes framework is one of the earliest formal theories of causal inference (Sjölander, 2012). It defines causal effects as the difference in potential outcomes under different levels of exposure or treatment (Rubin, 2005). This framework uses the language of potential outcomes to express causal effects in terms of joint distributions of potential outcomes represented as random variables. Causal assumptions in this framework are encoded as constraints on these distributions (Shpitser, 2012).

Potential outcomes can be categorized as *factual* (representing what actually occurred) or *counterfactual* (representing what would have occurred under different conditions). For example, if an individual took a medication and recovered, the factual outcome is "recovery," while the counterfactual outcome represents what would have happened if the medication had not been taken. Since counterfactual outcomes are inherently unobservable for an individual, estimating subject-specific causal effects is often impractical (Sjölander, 2012).

At the population level, however, counterfactual outcomes and causal effects can be estimated. Population-level causal effects contrast outcomes when everyone receives a treatment versus when no one does. Although only factual outcomes are observed, randomization allows for causal effect estimation under the Stable Unit Treatment Value Assumption (SUTVA) (Sjölander, 2012). Randomization ensures that potential outcomes are statistically independent of exposure, enabling identification of causal effects (Rubin, 2005). These principles are formally established in the literature (Rubin, 2005; Sjölander, 2012).

While the potential outcomes framework is widely used, it has limitations. Pearl has critiqued the framework for not providing systematic guidelines on which covariates to include for adjustment (Pearl, 1988). He warns that including all available covariates may inadvertently increase bias, highlighting the need for caution when selecting adjustment variables.

### B.2. Non-Parametric Structural Models (NPSEM)

The framework proposed by Pearl (Pearl, 2009a) is often celebrated for its coherence and robust formal foundations (Dawid, 2010). Pearl integrates principles from agency causality (focused on interventions), probabilistic graphical models (Dawid, 2010), and counterfactual reasoning (Sjölander, 2012). His approach balances the probabilistic view of causality from Bayesian models and the deterministic view from structural equation models (SEMs) common in econometrics and social sciences (Pearl, 2009a).

The NPSEM framework represents causal relationships us-

ing directed acyclic graphs (DAGs). A DAG $\mathcal{G} = (\mathbf{V}, \mathcal{E})$ consists of a set of variables $\mathbf{V}$ and directed edges $\mathcal{E}$ that encode causal dependencies. The structure ensures no cycles are formed. DAGs connect causal structure with joint probability distributions via the Markov condition, which states that each variable is conditionally independent of its non-descendants given its parents.

DAGs not only capture conditional independence but also distinguish causal from non-causal data-generating processes. If a variable $Y$ has an incoming edge from $X$, $X$ is a direct cause of $Y$. Indirect causation is mediated by intermediate variables; for instance, if $X$ influences $Y$ via $Z$, then $Z$ is a mediator. The NPSEM framework also provides criteria for determining the identifiability of causal quantities from observational data (Pearl, 2009a), making it a powerful tool for causal inference (Shpitser, 2012).

### B.3. Alternative Approaches

The sufficient cause framework views causation as a set of sufficient conditions leading to an event (VanderWeele, 2012; Mackie, 1965). Unlike the potential outcomes approach, which emphasizes causes, this framework focuses on effects (VanderWeele, 2012). Pearl extends this by proposing probabilistic notions of necessity and sufficiency (Pearl, 2009a).

The decision-theoretic approach incorporates stochastic counterfactuals to facilitate inference transportability between observational and experimental settings (Berzuini et al., 2012). This approach relaxes strong assumptions often required by potential outcomes (Dawid, 2015).

Finally, structural equation models (SEMs), rooted in deterministic relationships expressed through structural linear equations, remain widely used but are limited by their parametric assumptions and inability to model complex, non-linear causal relationships (Wright, 1921).

## C. Causality: Technical Preliminaries

### C.1. Causal Structures

Variables are represented by capital letters (e.g., $X$, $Y$), while specific values of variables are indicated using lowercase letters (e.g., $A = a$, $W = w$). Sets of variables and their values are denoted by bold capital letters (e.g., $\mathbf{V}$) and bold lowercase letters (e.g., $\mathbf{v}$), respectively.

A causal graph, denoted as $\mathcal{G} = (\mathbf{V}, \mathcal{E})$, is a Directed Acyclic Graph (DAG) consisting of a set of variables or nodes $\mathbf{V}$ and edges $\mathcal{E}$. Each edge $X \rightarrow Y$ signifies a causal relationship, meaning changes in $X$ directly influence $Y$. Importantly, altering $X$ impacts $Y$, but modifying $Y$ does not affect $X$.

Causal graphs include three foundational structures: **mediators**, **confounders**, and **colliders** (Pearl, 2009b), as illustrated in Figure 3:

- **Mediator**: A variable $W$ (Figure 3a) mediates the effect of $X$ on $Y$. For instance, $X \rightarrow W \rightarrow Y$ shows $X$'s influence on $Y$ through $W$. Mediators are also called chain structures.

- **Confounder**: A variable $C$ (Figure 3b) is a common cause of $X$ and $Y$, resulting in a non-causal correlation between them. While $X$ and $Y$ are correlated in this structure, $X$ does not directly cause $Y$.

- **Collider**: A variable $Z$ (Figure 3c) is influenced by $X$ and $Y$. Unlike the other structures, $X$ and $Y$ are uncorrelated unless conditioned on $Z$. Colliders are also known as v-structures.

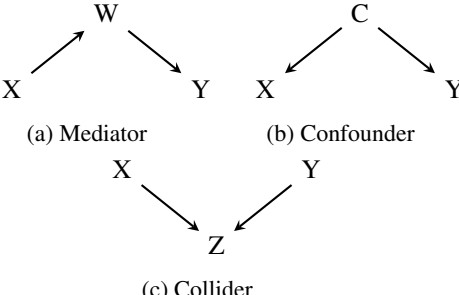

(a) Mediator      (b) Confounder

(c) Collider

Figure 3: Basic structures of causal graphs.

### Mediation Analysis

Causal relationships often involve multiple pathways, requiring mediation analysis to distinguish between them. For example, the causal effect between $X$ and $Y$ can be decomposed into:

- **Direct effect**: The path $X \rightarrow Y$.

- **Indirect effects**: Paths such as $X \rightarrow R \rightarrow Y$ and $X \rightarrow E \rightarrow Y$.

- **Path-specific effects**: Effects through a specific path, such as $X \rightarrow E \rightarrow Y$.

This decomposition is critical for fairness. A **direct effect** of $X$ on $Y$ is typically considered unfair when $X$ is a sensitive attribute (e.g., gender or race). In contrast, indirect effects may be fair or unfair, depending on the mediator. For example:

- An indirect effect through a discriminatory variable ($R$) is unfair.

- An indirect effect through an acceptable explanatory variable ($E$) is considered fair.

A variable is deemed a proxy (e.g., $R$) if it serves as a substitute for $X$ and leads to the same discriminatory outcome. Determining whether a variable is a proxy or an acceptable mediator often requires domain expertise.

### C.2. Causal Fairness Notions

Causal fairness aims to ensure that sensitive attributes, such as race or gender, do not unfairly influence outcomes. Below, we describe key causal fairness notions and their formal definitions.

#### C.2.1. TOTAL EFFECT (TE)

Total Effect (TE) (Pearl, 2009b) is a causal fairness notion that quantifies the overall effect of a sensitive attribute $X$ on an outcome $Y$. Formally, TE is defined as:

$$TE_{x_1,x_0}(y) = P(Y = y \mid do(X = x_1)) - P(Y = y \mid do(X = x_0)), \quad (1)$$

where $do(X = x)$ denotes an intervention that sets $X$ to $x$. TE measures the causal impact of changing $X$ from $x_0$ to $x_1$ on $Y$ across all causal paths connecting $X$ to $Y$.

#### C.2.2. MEDIATION ANALYSIS: NDE, NIE, AND PSE

Mediation analysis decomposes the causal effect of $X$ on $Y$ into direct and indirect effects. This is essential for identifying the pathways through which $X$ influences $Y$.

**Natural Direct Effect (NDE)** (Pearl, 2001): The NDE quantifies the direct effect of $X$ on $Y$, bypassing any mediators. For a binary variable $X$ with values $x_0$ and $x_1$, the NDE is:

$$NDE_{x_1,x_0}(y) = P(y_{x_1,\mathbf{z}_{x_0}}) - P(y_{x_0}), \quad (2)$$

where $\mathbf{Z}$ represents the set of mediator variables, and $P(y_{x_1,\mathbf{z}_{x_0}})$ is the probability of $Y = y$ if $X$ is set to $x_1$ while the mediators are set to values they would take under $X = x_0$.

**Natural Indirect Effect (NIE)** (Pearl, 2001): The NIE captures the influence of $X$ on $Y$ through mediators. It is given by:

$$NIE_{x_1,x_0}(y) = P(y_{x_0,\mathbf{z}_{x_1}}) - P(y_{x_0}), \quad (3)$$

where $P(y_{x_0,\mathbf{z}_{x_1}})$ represents the probability of $Y = y$ when $X = x_0$ but mediators take values they would under $X = x_1$.

**Path-Specific Effect (PSE)** (Pearl, 2009b; Chiappa, 2019; Wu et al., 2019): The PSE isolates the causal effect of $X$ on $Y$ transmitted through a specific path or set of paths $\pi$. Formally, it is defined as:

$$PSE_{x_1,x_0}^{\pi}(y) = P(y_{x_1|_{\pi}, x_0|_{\overline{\pi}}}) - P(y_{x_0}), \quad (4)$$

where $P(y_{x_1|_{\pi}, x_0|_{\overline{\pi}}})$ is the probability of $Y = y$ if $X = x_1$ along path $\pi$, while other paths ($\overline{\pi}$) remain unaffected by the intervention.

#### C.2.3. NO UNRESOLVED DISCRIMINATION

*No unresolved discrimination* (Kilbertus et al., 2017) requires that any causal effect of a sensitive attribute $X$ on an outcome $Y$ occurs only through resolving (explanatory) variables. A resolving variable, such as education level, reflects a non-discriminatory influence of $X$ on $Y$. The criterion prohibits direct and proxy effects of $X$ on $Y$.

#### C.2.4. NO PROXY DISCRIMINATION

*No proxy discrimination* (Kilbertus et al., 2017) ensures that decisions are not influenced by variables $R$ that act as proxies for sensitive attributes $X$. Proxy discrimination is absent if:

$$P(Y \mid do(R = r)) = P(Y \mid do(R = r')), \quad \forall r, r' \in \mathrm{dom}(R). \quad (5)$$

This guarantees that changes in $R$ do not affect the outcome $Y$ if $R$ is a proxy for $X$.

#### C.2.5. COUNTERFACTUAL FAIRNESS

*Counterfactual fairness* (Kusner et al., 2017) requires that the outcome $Y$ for an individual remains the same in both factual and counterfactual scenarios. Formally, counterfactual fairness holds if:

$$P(y_{x_1} \mid \mathbf{V} = \mathbf{v}, X = x_0) = P(y_{x_0} \mid \mathbf{V} = \mathbf{v}, X = x_0), \quad (6)$$

where $\mathbf{V}$ represents all other variables in the causal graph. This definition ensures fairness at the individual level by requiring that the sensitive attribute $X$ does not influence $Y$ in any hypothetical scenario.

## D. Causality and ML

The use of causality in AI falls mainly into one of two categories. The first approach is to employ artificial intelligence to enhance the qualitative discovery and/or quantification of causal connections from the data. The second one is to use causal tools to improve Machine Learning (ML) predictions. Next, we elaborate on both of these methods to combine causality and ML.

### D.1. ML for causality

**Causal Discovery** Most of the techniques for obtaining causal quantities rely on knowing the causal structure of the

data. It was previously assumed to be provided by experts. Recent advances in causal discovery offer algorithmic tools for recovering causal graphs from observational data. The basis for causal discovery is the probabilistic and graphical concepts of causality (Dawid, 2010). Two main groups of causal discovery algorithms can be distinguished based on their attempt to identify conditional or unconditional (including pairwise) independencies in the distribution from which the observational data is generated. The first category includes constraints and score-based algorithms such as PC (Le et al., 2016), FCI (Spirtes et al., 2000), and GES (Hauser & Bühlmann, 2012). They usually produce a partially oriented causal graph. The second category consists of algorithms based on causal asymmetries such as LiNGAM (Shimizu et al., 2006), and PNL (Zhang & Hyvarinen, 2012). The algorithms based on Kolmogorov's (algorithmic) complexity assume that if knowing the shortest compression of one variable does not reveal the shorter compression of the other, two variables are considered independent (Janzing & Schölkopf, 2010; Schölkopf, 2022). The summary of the principles and performance for pairwise causal discovery is provided by Mooij et al. (Mooij et al., 2016). If the assumptions of the algorithms are satisfied, they are capable of identifying a unique causal graph or a causal direction between the two variables.

**ML Tools for Causal Inference** Supervised or semi-supervised machine learning methods can be used to estimate causal quantities from the data or for variable selection in situations with a high number of covariates (Kreif & DiazOrdaz, 2019; Aoki & Ester, 2022). ML algorithms such as, for example, logistic regression, bagging, random forest, and others, can be beneficial in estimating propensity scores used to estimate causal effects in the potential outcome framework (Lee et al., 2010; Tu, 2019).

**LLMs for Causal Discovery** The recent advancements in large language models (LLMs) have inspired their use in causal discovery (Kıcıman et al., 2023; Kasetty et al., 2024; Vashishtha et al., 2023; AI4Science & Quantum, 2023; Abdulaal et al., 2023; Khatibi et al., 2024). Most of the above methods involve the refinement of the statistically inferred causal graph by LLM. However, emerging research shows that, LLMs excel at synthesizing vast amounts of heterogeneous knowledge, making them well-suited for tasks that require the integration of diverse datasets, such as constructing full causal graphs based on scientific literature in diverse domains (Sheth et al., 2024; Afonja et al., 2024).

### D.2. Causality for ML

One of the main arguments that motivated the use of causality for machine learning is that causal modeling can lead to more invariant or robust models (Schölkopf, 2022). The problem of overfitting and vulnerability to a domain shift is a known problem in ML. It is intuitive that learning the correlation between two phenomena, for example, rain and umbrellas, will not help to predict rain in situations where people prefer raincoats instead of umbrellas. A causal understanding of phenomena is more general to multiple circumstances. Following Pearl, "...we may as well view our unsatiated quest for understanding how data is generated or how things work as a quest to acquire the ability to make predictions under a wider range of circumstances, including circumstances in which things are taken apart, reconfigured, or undergo spontaneous change" (Pearl, 2009a). One of the methods to combine the ML model with the causal approach is to incorporate causal knowledge (usually in the form of a complete or partial causal graph) in the learning process (Berrevoets et al., 2023; 2022). Causal representation learning is an attempt to combine latent variables derived from unstructured data and causal structure to arrive at a more invariant or fair model (Schölkopf et al., 2021; Mitrovic et al., 2020; Schölkopf, 2022; Wang et al., 2022). The causal structure can also be used for feature selection, assuming that it is known. Models based on direct causes to predict the outcome are considered more robust (Tople et al., 2020).

