# OpenReview forum: "Position: Causality Is Key to Understand and Balance Multiple Goals in Trustworthy ML and Foundation Models"
_ICML.cc/2025/Position_Paper_Track — Submitted to ICML 2025 Position Paper Track_

### Official Review · Reviewer_3EZn · 2025-03-14

**Significance:** 4
**Argument Clarity:** 3
**Rating:** 4
**Confidence:** 3

**Questions:**

- Given the difficulty of obtaining high-quality causal graphs, how do you propose to scale causal integration in machine learning systems where ground truth causal structures are unknown?
- While causality can help disentangle bias sources, do you foresee scenarios where causal fairness approaches still fail to reconcile competing fairness notions?
- Is it possible that sufficiently large models implicitly learn causal structures without explicit causal modeling?

**Discussion Potential:**

3

**Paper Summary:**

This paper argues that causality is crucial for balancing multiple competing objectives in trustworthy machine learning and foundation models. These objectives include fairness, privacy, robustness, accuracy, and explainability, which are often addressed in isolation, leading to conflicts. The authors propose integrating causal methods into trustworthy machine learning to navigate these trade-offs systematically. They review existing applications of causality in trustworthy machine learning, suggest strategies for incorporating causal reasoning into foundation models, and discuss challenges and future research directions.

**Position:**

Yes

**Position In Title:**

Yes

**Related Work:**

4

**Strengths And Weaknesses:**

**Strengths:**
- This paper brings important discussion on causality in trustworthy machine learning, which aligns with increasing regulatory and ethical concerns with generative models.
- This paper effectively addresses key trustworthy ML objectives, such as fairness, privacy, robustness, explainability, and their trade-offs. It also provides an extensive literature review to support its arguments.
- This paper provides a broad framework for integrating causal reasoning across multiple dimensions, it also provides practical consideration for causal integration.

**Weaknesses:**
- There lacks discussion on computational cost of causal methods. While causality offers interpretability and robustness, it may increase computational cost, decrease scalability and efficiency.
- The paper acknowledges the difficulty of acquiring high-quality causal knowledge, but it does not sufficiently discuss practical strategies to overcome this limitation.
- The discussion on fairness highlights causal approaches like counterfactual fairness, but it does not fully address how different fairness might still conflict, even within a causal framework.

**Support:**

3

---

> ### Author Rebuttal · Authors · 2025-04-01
>
> We would like to thank the reviewer for their thoughtful feedback, highlighting the importance of our paper's discussion on causality in trustworthy machine learning, and its alignment with key objectives like fairness, privacy, robustness, and explainability.
>
> > Given the difficulty of obtaining high-quality causal graphs, how do you propose to scale causal integration in machine learning systems where ground truth causal structures are unknown?
>
> We thank the reviewer for the question. In the paper, we discuss several methods for scaling causal knowledge integration. On approach is based recent advancements in leveraging LLM-based causal discovery methods, where models either take predefined variable names as input or extract causal relationships directly from textual data. As mentioned in L378-380, this allows for the automated priors for causal discover or extraction of causal structures in domains where expert-defined causal graphs infeasible to construct manually. We also discuss causal knowledge integration in foundation models without the use of fully specified DAGs.
>
> >The discussion on fairness highlights causal approaches like counterfactual fairness, but it does not fully address how different fairness might still conflict, even within a causal framework. While causality can help disentangle bias sources, do you foresee scenarios where causal fairness approaches still fail to reconcile competing fairness notions?
>
> We thank the reviewer for this insightful question. As noted in the paper, recent work (e.g., Rahmattalabi & Xiang, 2022) has shown that causal fairness notions may avoid some fundamental incompatibility results present for statistical definitions of fairness, because it helps to disentangle different sources of bias. That said, we acknowledge that some fairness conflicts stem from deeper normative or value-based disagreements—for example, a causal relationship may exist, but relying on it in decision-making may still be viewed as unfair from an ethical or legal standpoint. In such cases, causality does not offer a definitive solution but provides a framework to surface these assumptions transparently, enabling more informed discussions that go beyond technical considerations.
>
> > Is it possible that sufficiently large models implicitly learn causal structures without explicit causal modeling?
>
> Indeed, as noted in the paper, recent work by Richens & Everitt (2024) provides theoretical support for the idea that robust agents implicitly learn causal models when exposed to sufficiently diverse data. However, we argue, that taking explicit steps for enhancing causal modeling improves explainability and gives better guarantees for achieving causality.
>
> >There lacks discussion on computational cost of causal methods. While causality offers interpretability and robustness, it may increase computational cost, decrease scalability and efficiency.
>
> That is a great point, we will mention the role of computational costs more explicitly. However, not all causal methods increase computational costs. For example, including the causal restrictions on ML models may actually decrease computation by narrowing down the search space.
>
> >The paper acknowledges the difficulty of acquiring high-quality causal knowledge, but it does not sufficiently discuss practical strategies to overcome this limitation.
>
> While causal graphs in the form of DAGs can be harder to obtain, the vast corpus of causal knowledge is available from scientific research in multiple fields. In our position paper we urge to focus on hybrid approaches and finding better ways of integrating causal knowledge. We also discuss several existing approaches that could point in the right direction.

---

### Official Review · Reviewer_1v5t · 2025-03-14

**Significance:** 4
**Argument Clarity:** 3
**Rating:** 2
**Confidence:** 5

**Questions:**

1) In discussing how causal approaches might scale to foundation models, the paper acknowledges “the difficulty of validating causal structures” given large, high-dimensional data. Do the authors have insights into what kind of techniques or verification steps shall be taken to validate causal structures in foundation models?

2) Regarding techniques such as "Causally Fair Language Models," is it feasible (and easy) to be generalized for a wide range of fairness issues, or are they limited to a very specific design target (e.g., demographic neutral)?

3) It is curious for the reviewer whether the term "accuracy" used throughout the manuscript needs to be re-defined differently between traditional ML models and causality models, especially with the author's discussion on "prediction accuracy vs. intervention accuracy"

**Discussion Potential:**

4

**Paper Summary:**

The paper advocates for integrating causal reasoning into ML and foundation models to balance multiple competing objectives of trustworthy AI, including fairness, privacy, robustness, accuracy, and explainability. The authors argue that current ML approaches often address these objectives in isolation, leading to conflicts and suboptimal solutions. As a fundamental solution, the paper states that causal inference can help reconcile these trade-offs by modeling data-generating processes and clarifying assumptions. For such purpose, the authors provide a review of the literature covering how causality has been integrated into ML and foundation model development. The paper then discusses potential applications of causality across different stages of foundation model development, from pre-training and fine-tuning to auditing and evaluation.

## update after rebuttal
While the author made extensive efforts in clarifying my concerns in their rebuttal, and some comments can be addressed through limited edits to the manuscript, my major concern about "There is a clear discontinuity between section 2.2 and section 3.2, which all investigated how causality could be integrated into ML/foundation models…More generally, the paper lacks a holistic synthesis of literature…" is difficult to be addressed. The " Specifically, we will..." action items proposed by the authors in their rebuttal will introduce too much edits to the manuscript which basically make it another version of the paper.

**Position:**

Yes

**Position In Title:**

Yes

**Related Work:**

2

**Strengths And Weaknesses:**

Strength:
The paper presents a timely argument for integrating causality to balance key trade-offs in trustworthy AI. The reviewer fully agrees that such a topic is important yet under-represented in the AI community. The paper performed a systematic examination of the tensions (e.g., Privacy vs. Accuracy) and proposed how causal methods could potentially solve them.

Weakness:
1) There is a clear discontinuity between section 2.2 and section 3.2, which all investigated how causality could be integrated into ML/foundation models. Without any explicit or implicit links between these two sections, even the discussion on such, such discontinuity makes the manuscript feel more like two loosely connected literature surveys than a cohesive narrative. A clearer roadmap or bridging discussion could show how the methods reviewed in Section 2.2 could transition into, or inform, the mehtods reviewed in Section 3.2, thereby providing a more unified structure.
2) More generally, the paper lacks a holistic synthesis of literature. Although it cites a range of studies applying causality to ML and foundation models, it often lists them back-to-back without explaining their conceptual links or contrasting their approaches. A more thorough synthesis would clarify how these existing works align with (or differ from) the paper’s overarching argument.
3) In multiple sections, the paper reiterates the same causal techniques, e.g., counterfactual methods and structural causal models, yet does not clearly delineate how each mention extends or differs from the others. This redundancy makes it difficult for the reader to see which points are original contributions from this work vs. which are just reemphasizing standard practices. For example, the discussion on counterfactual fairness appears in several places, but there is no clear distinction between each appearance. Such repeated content without additional nuance or comparative analysis can obscure the novel insights that should stand out from the writing.

Technically, the position paper shall put more discussion on the limitation of causal inference from a practical point of view, which is a common and serious challenge in implementation. Specifically, many causal methods require strong assumptions about the data-generating process (e.g., availability of causal graphs which are covered in the paper, identifiable interventions, clear contextual variables, etc.), which are often unrealistic in real-world applications. Additionally, the paper does not sufficiently address the scalability concerns that arise when integrating causality (mainly computationally), an issue that becomes especially pressing with the ever-growing sizes of foundation models.

**Support:**

2

---

> ### Author Rebuttal · Authors · 2025-04-01
>
> We thank the reviewer for their encouraging feedback, particularly for recognizing our central argument that causality helps navigate trade-offs in trustworthy AI. We are glad the reviewer finds this topic timely and underrepresented, and we appreciate the acknowledgment of our effort to explore tensions in trustworthy AI and propose causal solutions.
>
> > In discussing how causal approaches might scale to foundation models, the paper acknowledges “the difficulty of validating causal structures” given large, high-dimensional data…
>
>  In the context of mechanistic interpretability, techniques such as causal scrubbing and causal abstraction have been proposed to validate causal hypotheses through targeted interventions within the model’s internal representations. These methods offer promising first steps, however, we fully acknowledge that further theoretical and methodological advances are needed and urge for it in our position paper.
>
> >Regarding techniques such as "Causally Fair Language Models," is it feasible (and easy) to be generalized for a wide range of fairness issues…
>
> We thank the reviewer for pointing out the need to specify the fairness issues discussed. In our discussion, the focus is on fair representation, which can involve tensions with historical accuracy, e.g., in the known case of generating black Founding Fathers. In such cases, causal reasoning helps navigate these tensions by applying context-aware restrictions.
> However, in principle, causal approach to fairness can be extended to broader fairness criteria by conditioning on different causal factors relevant to the fairness definition at hand.
>
> > It is curious for the reviewer whether the term "accuracy" used throughout the manuscript needs to be re-defined differently between traditional ML models and causality models, especially with the author's discussion on "prediction accuracy vs. intervention accuracy"
>
> Thank you for pointing out. In this paper, “accuracy” mostly refers to prediction accuracy—as in standard supervised learning contexts. However, we make a distinction between prediction accuracy and intervention accuracy (how well it predicts the effect of interventions), which is important in certain domains and therefore requires causal models. Additionally, in generative AI contexts, accuracy can refer to distributional faithfulness, factual correctness (especially in LLMs), or utility. We are happy to revise the manuscript to clarify these distinctions.
>
> > There is a clear discontinuity between section 2.2 and section 3.2, which all investigated how causality could be integrated into ML/foundation models…More generally, the paper lacks a holistic synthesis of literature…
>
> We thank the reviewer for the thoughtful and constructive suggestion. We acknowledge that the transition between Section 2.2 and Section 3.2 could benefit from a clearer narrative bridge. The perceived discontinuity arises, in part, from the differing levels of maturity between causal methods in traditional ML and those being developed for foundation models, as well as architectural and training differences. While causal ML has a well-established foundation with a range of formal methods, integrating causality into foundation models is still an emerging area, with distinct architectural and operational challenges.
> To address this, we will revise the manuscript to include an explicit roadmap and bridging discussion that connects these two sections. Specifically, we will:
> - Add a transitional paragraph at the end of Section 2.2 or beginning of Section 3.2 to explain how the core causal ML principles extend into the foundation models as well as important differences;
>
> - Highlight conceptual mappings, such as how invariant feature learning is related to causal-invariant fine-tuning, and how disentangled VAEs relate to causal representation learning in pre-training.
>
> > In multiple sections, the paper reiterates the same causal techniques, e.g., counterfactual methods and structural causal models...
>
> We thank for pointing this out.  We use counterfactual fairness twice for ML models and LLMs, to note, that it is applicable in both cases. The other mentions of “counterfactuals” refer to counterfactual explanations or counterfactual data augmentation. We are sorry about the confusion and will delineate it more clearly.
>
> >Technically, the position paper shall put more discussion on the limitation of causal inference from a practical point of view...
>
> We thank the reviewer for highlighting the importance of discussing the practical limitations of causal inference. In Section 4 of the paper, we do discuss several of these challenges, including the identifiability assumptions. That said, our position paper adopts a broad view of causality in ML and also highlights less restrictive approaches that help to guide model towards causal representations, such as invariant feature learning, sparse autoencoders, counterfactual data augmentation and use of partial causal knowledge.

---

> > ### Comment · Reviewer_1v5t · 2025-04-05
> >
> > While the author made extensive efforts in clarifying the reviewer's concerns, and some comments can be addressed through limited edits to the manuscript. The major concern from the reviewer "There is a clear discontinuity between section 2.2 and section 3.2, which all investigated how causality could be integrated into ML/foundation models…More generally, the paper lacks a holistic synthesis of literature…" is difficult to be addressed. The " Specifically, we will..." action items will introduce too much edits to the manuscript which basically make it another version of the paper. So I'll keep my score.

---

### Official Review · Reviewer_Dk6j · 2025-03-15

**Significance:** 2
**Argument Clarity:** 3
**Rating:** 2
**Confidence:** 3

**Questions:**

Can the paper include empirical studies or data to support its theoretical claims, and discussion on real-world implementations?

**Discussion Potential:**

2

**Paper Summary:**

The paper advocates the position that emphasizes integration of causal reasoning into machine learning to address the inherent trade-offs between key dimensions of trustworthy AI such as fairness, privacy, robustness, explainability, and accuracy. The authors state that causality can provide a structured approach to navigating these trade-offs by explicitly modeling data-generating processes, thereby enhancing the reliability and interpretability of ML systems. They discuss practical implementations of causal methods in both traditional machine learning and foundation models, suggesting future directions for incorporating causal reasoning to improve AI systems' accountability and ethical soundness.

**Position:**

Yes

**Position In Title:**

Yes

**Related Work:**

3

**Strengths And Weaknesses:**

Strengths:
1. The paper effectively discusses how causality can aid in balancing multiple components of trustworthy AI, providing a fresh perspective compared to traditional correlation-based approaches.
4. The paper uses examples and potential case studies to illustrate how causal methods can be implemented.
5. The paper clearly outlines the challenges and limitations of implementing causality in AI systems, which prepares the field for addressing these issues in future research.

Weaknesses:
1. My biggest concern is that the paper is largely theoretical and lacks empirical results to substantiate the claims about the effectiveness of causal approaches in improving AI systems.
2. The paper assumes availability of causal knowledge or the ability to generate it, which might not be feasible in all domains or datasets. Also, the integration of causality as described can be complex and may require substantial changes to existing ML pipelines, which might limit its practicality for some users.
3. Potential overemphasis on causality: while causality is crucial, the paper might benefit from a more balanced discussion that also considers the roles of other approaches in achieving trustworthy AI.
4. The paper discusses scalable methods but does not provide concrete solutions for implementing these at the scale required by large models like GPT-4 or BERT.
5. The methods for causal discovery can be biased by the data and assumptions used, which is not thoroughly addressed. Also, there is limited detailed discussion on specific methodologies to counteract these biases effectively across different AI systems.

**Support:**

1

---

> ### Author Rebuttal · Authors · 2025-04-01
>
> We thank the reviewer for their thoughtful  feedback. We’re grateful for the recognition of our efforts to present a fresh perspective on trustworthy AI through the lens of causality.  We're glad that the use of illustrative examples helped convey how causal methods can be practically implemented and appreciate the acknowledgment of our discussion around the challenges and limitations.
>
> >Can the paper include empirical studies or data to support its theoretical claims, and discussion on real-world implementations?
> My biggest concern is that the paper is largely theoretical and lacks empirical results to substantiate the claims about the effectiveness of causal approaches in improving AI systems.
>
> We appreciate the reviewer’s concerns regarding the paper’s primarily theoretical nature. However, we would like to clarify that this is a position paper, and as such, its purpose is to articulate a viewpoint and outline promising directions—rather than to propose or validate a specific model through empirical results.
> That said, we do ground our position in a broad range of existing empirical studies and highlight real-world applications and use cases drawn from existing literature across different domains and model types. Many of the works we cite provide strong evidence of how causal approaches have led to measurable improvements in areas such as fairness, robustness, and generalization.
>
> > The paper assumes availability of causal knowledge or the ability to generate it, which might not be feasible in all domains or datasets. Also, the integration of causality as described can be complex and may require substantial changes to existing ML pipelines, which might limit its practicality for some users.
>
> We agree that causal knowledge is not always available, and causal discovery methods can be sensitive to data and assumptions. We explicitly discuss these challenges in Section 4 (Challenges and Opportunities) and offer several paths forward, including:
>
> - Hybrid approaches combining causal discovery with expert or LLM priors for scalable discovery;
>
> - Emphasis on using or augmenting with causal data rather than requiring complete causal graphs;
>
> - Integration strategies based on the stage of existing pipeline.
>
>
> >Potential overemphasis on causality: while causality is crucial, the paper might benefit from a more balanced discussion that also considers the roles of other approaches in achieving trustworthy AI.
>
> While the position of the paper is the potential of causal reasoning in addressing core trade-offs in trustworthy AI, we do not suggest it is the only approach. In fact, Section 5 provides alternative perspectives.
> However, to clarify the intent of the paper, we would like to highlight that the primary purpose of a position paper is to present a compelling viewpoint that warrants greater attention, regardless of whether every reader fully agrees with it. In line with the CfP's intent, we aim to emphasize the value of causality as a pivotal tool in addressing trade-offs in trustworthy AI, given the current gaps in the literature and the potential of causal methods to provide a structured, principled approach to these challenges. We would be happy to include more perspectives, but we also feel that this paper is meant to spark a conversation and advocate for the role of causality within this important discourse, not to cover the entire landscape of trustworthy AI approaches comprehensively.
> We hope this helps clarify our position and approach in the paper, and we are open to further refining the text to better balance the perspectives.
>
>
> > The paper discusses scalable methods but does not provide concrete solutions for implementing these at the scale required by large models like GPT-4 or BERT.
>
> We discuss concrete directions for integrating causal reasoning into foundation models (Section 3.2), covering practical implementations across:
>  - Pre-training (e.g., causal data augmentation, causal embedding, entity-level interventions);
>
>  - Post-training (e.g., counterfactual fine-tuning, causal RLHF);
>
>  - Auditing and evaluation (e.g., causal attribution).
>
> These examples draw from current literature applying causality in models like BERT and GPT-style architectures, showing that while challenges remain, the approach is feasible and already being explored at scale.
>
> > The methods for causal discovery can be biased by the data and assumptions used, which is not thoroughly addressed. Also, there is limited detailed discussion on specific methodologies to counteract these biases effectively across different AI systems.
>
> We acknowledge the limitations of causal discovery methods and highlight this explicitly in Section 4. We also discuss how combining discovery with priors from expert knowledge can mitigate such biases. Importantly, we also note that many proposed causal integration strategies do not rely on causal discovery at all, instead leveraging known or labeled causal information.

---

### Official Review · Reviewer_djxQ · 2025-03-19

**Significance:** 3
**Argument Clarity:** 3
**Rating:** 1
**Confidence:** 4

**Questions:**

What are the key challenges in effectively integrating causality in ML models? If the causal structure is not available and the inferred causal structures are not accurate, what are the effective ways to utilize the inaccurate causality in the ML models?

**Discussion Potential:**

2

**Paper Summary:**

The paper argues for integrating causal approaches in future ML and foundation models to achieve trustworthy solutions. It first discusses the necessity of causal components in all aspects of trustworthiness such as fairness, robustness, explainability, privacy etc, and then surveys existing approaches of integrating causality into ML models.

## update after rebuttal
The rebuttal does not fully address my concern. It repeats the obvious claims already made in the paper, but avoids answering the more important question on how to effectively incorporate causality into trustworthy ML and foundation models.

**Position:**

Yes

**Position In Title:**

Yes

**Related Work:**

3

**Strengths And Weaknesses:**

Strengths:
1. The position statement is clearly articulated and well-argued
2. Some existing work along the causal ML models have been discussed

Weakness:
1. It is clearly important to integrate causality in ML models. But the main challenge is how the model will be developed in a way so that we can achieve fairness, robustness, explainability, privacy etc. Most existing work have not been successful in achieving the goal. The focus of the paper should be discussing the main challenges in causal ML models and providing clear directions to address these challenges.

**Support:**

3

---

> ### Author Rebuttal · Authors · 2025-04-01
>
> We thank the reviewer for confirming the importance of causality in ML models and need for effective methods for integrating it. The answers to the questions can be found below:
>
> >What are the key challenges in effectively integrating causality in ML models? If the causal structure is not available and the inferred causal structures are not accurate, what are the effective ways to utilize the inaccurate causality in the ML models?
>
> We appreciate the question and agree that integrating causality into ML models presents several important challenges. We discuss these in Section 4. To address some of the difficulties we are  taking a broad perspective on causal integration, relying not solely on full DAGs, but augmenting with causal data from experiments, or data with generated counterfactuals, as well as constraining with partial causal graphs.
>
> > It is clearly important to integrate causality in ML models. But the main challenge is how the model will be developed in a way so that we can achieve fairness, robustness, explainability, privacy etc. Most existing work have not been successful in achieving the goal. The focus of the paper should be discussing the main challenges in causal ML models and providing clear directions to address these challenges.
>
> Thank you for the feedback. We would like to clarify that the main goal of the position paper is not to propose a specific causal ML model, but rather to argue for a broader, principled integration of causality into the development of trustworthy ML systems. Our position emphasizes that causality is not just a tool for one specific goal (e.g., fairness), but a foundational perspective that can help resolve interacting trade-offs across fairness, robustness, explainability, privacy, and more.
> We do address the challenges in Section 4 and outline a number of directions that could help overcome them.
> In addition, we draw on a rich body of literature that conceptually softens or resolves key trade-offs—e.g., between fairness and accuracy (Chiappa et al., 2019)—through causal reasoning. Based on this, we argue that explicitly addressing these trade-offs through causal lenses is not only possible, but necessary for the development of trustworthy ML.
> That said, we appreciate the suggestion to expand on the challenges and are happy to do so.

---

### Decision · Program_Chairs · 2025-04-29

**Decision:**

Reject

**Comment:**

Due to constraints on conference capacity, we had to reject some papers despite a favorable recommendation.  The area chair's original meta-review (copied below) is a testament to the strengths of your paper.

---
Overall, most reviewers found the paper relevant and timely. I agree with the pushback by the authors that this is a position paper, and as such, it only needs to argue its position convincingly, not provide a technical solution to a technical problem; this position is for the authors to choose, and while there are other interesting problems ("how exactly are we going to deploy these causal models?"), I feel they are out-of-scope of the current position. I disagree with the claims that the stance is obvious---if it was, there would not be such a huge gap in the literature when it comes to causality in trustworthy AI. I also see that much of the comments about challenges are actually the object of Section 4 and in part addressed by the authors.

While it seems that Section 4 could be improved to discuss more challenges highlighted by the reviewers, I have decided to give this paper a weak accept to acknowledge the contributions, the careful response by the authors that I believe counters many of the points made in the review, and the strong support from reviewer 3EZn.

That said, I am also flagging this paper as needing further discussion with the chairs.